# Effect of 12-Week Daily Intake of the High-Lycopene Tomato (*Solanum Lycopersicum*), a Variety Named “PR-7”, on Lipid Metabolism: A Randomized, Double-Blind, Placebo-Controlled, Parallel-Group Study

**DOI:** 10.3390/nu11051177

**Published:** 2019-05-25

**Authors:** Mie Nishimura, Naoki Tominaga, Yuko Ishikawa-Takano, Mari Maeda-Yamamoto, Jun Nishihira

**Affiliations:** 1Department of Medical Management and Informatics, Hokkaido Information University, Ebetsu, Hokkaido 069-8585, Japan; mnishimura@do-johodai.ac.jp; 2Plant Breeding & Experiment Station, Takii & Co., Ltd., Konan, Shiga 520-3231, Japan; naoki-tominaga@takii.co.jp; 3Functional Food Factor Laboratory, Food Function Division, National Food Research Institute, National Agriculture and Food Research Organization (NARO), Tsukuba, Ibaraki 305-8642, Japan; yuko@affrc.go.jp; 4National Agriculture and Food Research Organization (NARO), Tsukuba, Ibaraki 305-8517, Japan; marimy@affrc.go.jp

**Keywords:** LDL-cholesterol, lipid metabolism, lycopene, PR-7, randomized controlled trial, semidried tomato

## Abstract

Tomato (*Solanum lycopersicum*) is a rich source of lycopene, a carotenoid that confers various positive biological effects such as improved lipid metabolism. Here, we conducted a randomized, double-blind, placebo-controlled, parallel-group comparative study to investigate the effects of regular and continuous intake of a new high-lycopene tomato, a variety named PR-7, for 12 weeks, based on 74 healthy Japanese subjects with low-density lipoprotein cholesterol (LDL-C) levels ≥120 to <160 mg/dL. The subjects were randomly assigned to either the high-lycopene tomato or placebo (lycopene-free tomato) group. Each subject in the high-lycopene group ingested 50 g of semidried PR-7 (lycopene, 22.0–27.8 mg/day) each day for 12 weeks, while subjects in the placebo group ingested placebo semidried tomato. Medical interviews were conducted, vital signs were monitored, body composition was determined, and blood and saliva samples were taken at weeks 0 (baseline), 4, 8, and 12. The primary outcome assessed was LDL-C. The intake of high-lycopene tomato increased lycopene levels in this group compared to levels in the placebo group (*p* < 0.001). In addition, high-lycopene tomato intake improved LDL-C (*p* = 0.027). The intake of high-lycopene tomato, PR-7, reduced LDL-C and was confirmed to be safe.

## 1. Introduction

Dyslipidemia is a major risk factor for coronary heart disease, and in Japan, the number of patients with this condition is increasing. Through a survey, the Ministry of Health, Labor and Welfare reported that 2.1 million patients received treatment for this condition in 2014. It is well known that dietary improvements are important for the prevention of dyslipidemia, and research on functional foods that affect lipid metabolism is receiving increased attention. In 2015, the system of “Foods with Function Claims” was established in Japan. As this system permits health claims for fresh vegetables, the research on and development of such vegetables containing highly functional components, also known as “functional vegetables”, is expected.

Lycopene, a carotenoid, has antioxidant effects and exhibits the highest physical quenching rate constant for singlet oxygen [1]. It has also been reported to inhibit the production of serum lipid peroxide and oxidize low-density lipoprotein (LDL) in a concentration-dependent manner [2]. A previous clinical trial with obese subjects suggested that the continuous ingestion of a diet containing high lycopene increased serum high-density lipoprotein (HDL)-2 and HDL-3, which are subtypes of HDL cholesterol (HDL-C) [3]. The biological mechanism was suggested to be through a reduction in 3-hydroxy-3-methyl-glutaryl-coenzyme A (HMG-CoA) reductase activity in the liver [4], activation of LDL-receptors [5], and increased expression of the ABCA1 transporter gene, which is the key component of HDL-C production [6]. An epidemiological survey also revealed that the continuous intake of lycopene contributes to the prevention of prostate cancer [7] and reduces risks related to the onset of cardiovascular disease [8].

Tomato (*Solanum lycopersicum*) is the main source of lycopene, possessing an amount that normally ranges from 3 to 5 mg lycopene per 100 g of raw tomato [9]. Takii & Co., Ltd. succeeded in breeding a high-lycopene tomato variety named “PR-7” (GOCHISO TOMATO), using a high-lycopene line selected from an edible tomato. PR-7 contains more lycopene than conventional tomatoes. This makes it possible to obtain the required amounts of lycopene from raw tomatoes to improve lipid metabolism. In addition, PR-7 contains high amounts of glutamic acid.

The effects of lycopene and tomato on lipid metabolism have been shown in previous clinical trials; however, almost all trials used tomato juice or tomato extract as the test food in double-blind, placebo-controlled studies [10,11,12]. This is because it is difficult to produce a placebo for raw tomato and to deliver such food products to subjects. We therefore opted to produce semidried tomato (semidried tomato contained 22% water content) and used lycopene-free tomato as the placebo. By using this test food in a double-blind, placebo-controlled study, we demonstrated that semidried tomato is similar to raw tomato and its lycopene-induced effect is the same as that of processed tomato such as juice or extract. 

In 2017, we conducted a pilot clinical trial called “Effect of Daily Ingestion of High-Lycopene Tomato PR-7 on Lipid Metabolism for 8 Weeks: A Randomized, Double-Blind, Placebo-Controlled, Parallel-Group (UMIN registration number: UMIN000026524)”. In this pilot study, continuous intake of 50 g of semidried PR-7 for 8 weeks tended to improve the lipid profile; therefore, we expected that the effect of ingesting high-lycopene tomato on lipid metabolism could be improved by extending the intake period.

Thus, we conducted a randomized, double-blind, placebo-controlled, parallel-group comparative study to evaluate the effects of continuous intake of PR-7 for 12 weeks using healthy subjects with LDL-C levels ≥120 to <160 mg/dL.

## 2. Materials and Methods 

### 2.1. Study Design

This placebo-controlled, randomized, double-blind, parallel-group comparative study was conducted at Hokkaido Information University, Health Information Science Research Center (Ebetsu, Hokkaido, Japan). The schedule for the trial is summarized in Table 1. Written informed consent was obtained from subjects on their first visit prior to being enrolled. The subjects ingested 50 g semidried tomato (active or placebo) every day for 12 weeks without cooking. Medical interviews, measurements of vital signs and body composition, saliva and urinary assessments, and hematological and biological assessments were conducted during the second (week 0; baseline), third (week 4), fourth (week 8), and fifth (week 12) visits. In addition, all subjects completed a Food Frequency Questionnaire Based on Food groups (FFQg) (Kenpakusha, Tokyo, Japan) at visits 2–4. During the entire course of this study, subjects were asked to maintain their daily activities including food consumption and exercise habits, to avoid any supplements, tomatoes, and processed foods containing tomatoes, and to avoid drinking any vegetable juices; the test food was the only tomato derivative that was allowed to be consumed. The subjects used a diary to record their daily activities, which was reviewed by a medical doctor or nurse at each visit.

The primary outcome assessed was the change in LDL-C. Secondary outcomes assessed were as follows: lipid profiles comprising total cholesterol (TC), HDL-C, triglycerides (TG), LDL-C/HDL-C ratio, and non-HDL; adiponectin; serum lycopene; serum β-carotene; malondialdehyde LDL-C (MDA-LDL); lectin-like oxidized LDL receptor-1 (LOX) index; soluble LOX LDL receptor-1 (sLOX-1); LOX-1 ligand containing apolipoprotein B (LAB); lipid peroxidases (LPO); saliva volume; salivary chromogranin A (CGA); results of the visual analog scale (VAS) questionnaire on fatigue and stress; the Profile of Mood States Second Edition (POMS-2) full-length version for adults. The efficacy of the active test food was evaluated at week 12 and its safety at weeks 4, 8, and 12.

### 2.2. Study Subjects

We screened 234 volunteers on their first visit, and all provided written informed consent to participate in this study. After screening, 100 healthy Japanese subjects were enrolled in the study (ages, ≥30 to <70 years; LDL-C, ≥120 to <160 mg/dL). The range of LDL-C was based on the “Guideline for Prevention of Arteriosclerosis Disease” published by the Japan Atherosclerosis Society. It defines the range of 120–139 mg/dL as representing borderline high cholesterol and the range of 140–159 mg/dL as indicating mild hypercholesterolemia. Inclusion and exclusion criteria are summarized in Table 2. The eligible subjects were randomly assigned to either the active test food (PR-7) or placebo food (lycopene-free tomato) groups stratified by sex, age, and LDL-C during the first visit. Assignments were computer generated based on stratified block randomization at a third-party data center (Media Educational Center, Hokkaido Institute of Information Technology, Ebetsu, Hokkaido, Japan). Medical doctors, nurses, clinical research coordinators, and statistical analysts were blinded to the assignment information during the trial period. The test foods were controlled by the food-controlled numbers printed on the food package. The information was disclosed only after all analytical data were collected and the subjects in the efficacy analysis and the method used for statistical analyses were finalized.

### 2.3. Preparation of the Test Food

The tomato, a variety named PR-7 (*Solanum lycopersicum*) that was bred by Takii & Co, Ltd (Kyoto, Japan) and harvested from Kasai (Hyogo, Japan) and Hokuto (Yamanshi, Japan), was used as the active test food; a lycopene-free tomato harvested from Ishii-cho (Tokushima, Japan) was used as the placebo. To prepare the semidried tomato: (1) raw tomatoes were disinfected; (2) the calyx on each tomato was removed prior to slicing; (3) slices were dried and vacuum-packed; (4) packed tomato was disinfected and refrigerated; (5) bacterial tests, physicochemical tests, and sensory inspections were conducted. Analyses of the nutrient composition of the active test food and the placebo were conducted using the methods established by the Japan Food Research Laboratories (Hokkaido, Japan) and are presented in Table 3. Lycopene was measured using nine randomly extracted samples. β-Carotene level was measured using high-performance liquid chromatography at Takii & Co., Ltd. The levels of calories and carbohydrates were different between test foods. However, compared to the daily intake of nutrition from the diet, this difference was insignificant; therefore, we considered that this would not affect our data. The active test food and placebo were prepared under strict quality-control protocols and were identical in appearance.

### 2.4. Physical, Hematological, Biological, Urinary, and Salivary Assessments

Blood was collected from subjects after a 12-h fast and used for the following hematological examinations: white blood cell (WBC), red blood cell (RBC), hemoglobin (Hb), hematocrit (Ht), and blood platelet (Plt) counts. Biological examinations included the following: liver function (aspartate aminotransferase [AST], alanine aminotransferase [ALT], gamma-glutamyl transpeptidase [γ-GTP], alkaline phosphatase [ALP], and lactate dehydrogenase [LDH]); renal function (blood urea nitrogen [BUN], creatinine [CRE], and uric acid [UA]); lipid profiles (TC, LDL-C, HLD-C, and TG); blood glucose profiles (fasting plasma glucose [FPG], hemoglobin [Hb]A1c, and homeostatic model assessment of insulin resistance [HOMA-IR]); adiponectin levels; serum carotenoids (lycopene and β-carotene); and oxidative markers (MDA-LDL, LOX index, sLOX-1, LAB, and LPO).

Saliva was collected using a Salivette^®^ Cotton Swab (Sarstedt K.K., Tokyo, Japan); subjects were asked to chew the swab for 60 s to stimulate salivation. Saliva volume and chromogranin A (CGA) were then assessed.

Urine was first collected in the morning, from which, pH, sugar, protein, occult blood, urobilinogen, and ketones were qualitatively assessed.

Blood, saliva, and urine tests were analyzed at Sapporo Clinical Laboratory, Inc. (Hokkaido, Japan). Measurements of serum carotenoid and oxidative markers were analyzed at NK Medico Co., Ltd (Tokyo, Japan). Body composition and blood pressure were measured using a Body Composition Analyzer DC-320 (Tanita Corp, Tokyo, Japan) and an Automatic Blood Pressure Monitor HEM-7080IC (Omron Co., Ltd., Kyoto, Japan), respectively.

### 2.5. VAS Questionnaire Assessing Fatigue, Stress, and Profile of Mood States

To evaluate the effects of high-lycopene tomato on fatigue and stress, subjects completed a VAS questionnaire with eight questions assessing fatigue and stress. Subjects were instructed to place an “X” along a 100-mm line to provide a rating from the worst to the best condition for each question based on their current health condition. The left end of the line (0 mm) was defined as the worst condition and the right end (100 mm), the best condition. The questionnaire results were assessed by evaluating the length from the beginning of the line on the left to the “X.” An increase in VAS score indicated an improvement in each symptom.

The POMS-2 questionnaire was used to evaluate the effects of high-lycopene tomato on mood [13]. Total Mood Disturbance (TMD) scores were assessed using the POMS-2 full version for adults, which comprised 65 questions (Success Bell, Tokyo, Japan). Subjects selected from five answers ranging from “not at all” to “quite a lot” based on their mood state over the previous week.

### 2.6. Food Frequency Questionnaire

The FFQg is a semi-quantitative dietary assessment and is used to estimate nutrient intake based on the subjects’ regular diet [14,15]. This questionnaire comprised 29 food groups and 10 types of cooking methods. For each question, subjects reported the weekly amount and frequency of food intake for the past month at each visit, from which regular and nutrient intakes (calories, protein, fat, carbohydrates, dietary fiber, and salt) were estimated.

### 2.7. Assessment of Safety

We further assessed the incidence of adverse effects or symptoms and abnormal changes in laboratory variables. The severity of adverse effects and their relation to the test food were classified according to protocol criteria set by the investigator. Laboratory variables were assessed according to the guideline of side effect criteria defined by the Japanese Society of Chemotherapy [16]. All adverse effects were reported as follows: symptoms, occurrence date, severity, relation to test food, continuation or discontinuation, treatment, and outcome. Adverse effects were monitored during the intervention for 12 weeks.

### 2.8. Ethics

The current clinical trial was conducted in compliance with the ethical guidelines on medical research with humans (Ministry of Education, Culture, Sports, Science and Technology, and Ministry of Health, Labor and Welfare) and the Declaration of Helsinki (revised by the Fortaleza General Meeting of the World Medical Association). The trial protocol was approved by the ethics committee of Hokkaido Information University (Ebetsu, Hokkaido, Japan; approved on 21 February, 2018; approval number: 2017-25). This trial is registered at www.umin.ac.jp/ctr/index.htm (registered on 29 March, 2018; registration number: UMIN000031975).

### 2.9. Statistical Analysis

Student’s *t*-tests were used to analyze primary and secondary outcomes, hematological and biological examinations, and food frequency questionnaire values by comparing the changes in subject values between the two groups. Changes in subject values were analyzed using repeated measures of analysis of variance between groups. For subject characteristics, t Fisher’s exact probability test was used for sex and the Mann–Whitney U-test was used for intake rate; Student’s *t*-tests were used for other subject characteristics. Based on subgroup analysis, we analyzed LDL-C in subjects for whom LDL-C levels were 120–139 mg/dL (borderline high cholesterol subjects) and 140–159 mg/dL (mild hypercholesterolemia subjects). All statistical analyses were performed using SPSS v. 25 (IBM Japan, Ltd., Tokyo, Japan), and *p* < 0.05 was considered statistically significant.

### 2.10. Sample Size

Prior to this study, we conducted a placebo-controlled, double-blind, parallel-group comparison test (registered at www.umin.ac.jp/ctr/index.htm; registration number, UMIN000026524; date of registration, 13 March, 2017) over an 8-week intake period with the same dosages used in this trial. Based on preliminary data, the sample size was calculated to detect an intergroup difference of 8.70 with respect to changes in LDL-C from baseline to week 8 (SD = 13.7) and an effect size of 0.63 using a two-sided paired *t*-test with a statistical power of 80% and an α of 5%. These results indicated that a sample size of 80 (40 in each group) was necessary. Assuming a 20% loss in follow-up, 100 subjects (50 in each group) were enrolled.

## 3. Results

### 3.1. Subject Dropouts and Characteristics

Subject involvement throughout the trial period is presented in Figure 1. Subjects who provided informed consent (*n* = 234) were assessed for eligibility, and after the screening, 100 subjects were enrolled in this study. All enrolled subjects were randomized into one of the two intervention groups (placebo group, *n* = 50; active test food group, *n* = 50). Prior to trial initiation, two subjects dropped out because of personal reasons, one subject dropped out because of difficulty ingesting the test food, three subjects dropped out because of personal reasons after the study began, and one subject dropped out after beginning other medical treatments. Finally, 93 subjects completed this trial—47 in the active test food group and 46 in the placebo group. There was no reason to stop this study in the middle, and we did not conduct an interim analysis; therefore, the study was completed as planned on 3 September 2018. Ninety-eight subjects, excluding the two subjects who dropped out because of personal reasons before the start of the trial, were included in the safety analysis. We excluded 19 subjects in the efficacy analysis because of abnormal value variations from mild sickness or disordered lifestyle (*n* = 5), compliance problems (*n* = 3), missing primary outcomes due to absence (*n* = 1), not being accustomed to eating raw tomatoes (*n* = 6), and regularly drinking tomato juice (*n* = 4). The efficacy analysis comprised 33 subjects in the active test food group and 41 in the placebo group. Sex ratio, mean age, height, body mass index (BMI), LDL-C at visit 1, and the intake rate for each group are presented in Table 4. These characteristics did not significantly differ between the two groups, confirming the appropriate allocation of the number of subjects to each group.

### 3.2. Effect of High-Lycopene Tomato on LDL-C

The intake of the active test food significantly improved LDL-C in this group compared to that in the placebo group (ΔWeek 12: placebo, 4.1 ± 15.7 mg/dL; high-lycopene tomato, −3.7 ± 13.8 mg/dL; *p* = 0.027; Figure 2, Table 5). Based on a subgroup analysis, LDL-C was significantly decreased at week 12 (ΔWeek 12: placebo, 4.3 ± 15.1 mg/dL; high-lycopene tomato, −5.1 ± 9.5 mg/dL; *p* = 0.030) following the ingestion of high-lycopene tomato in subjects with LDL-C ranging from 120–139 mg/dL (Table 5). In subjects with LDL-C ranging from 140–159 mg/dL, LDL-C decreased at week 12 but this was not significant (Table 5).

### 3.3. Effect of High-Lycopene Tomato on Lipid Profile and Adiponectin Levels

The effects of high-lycopene tomato on lipid profiles comprising TC, LDL-C, HDL-C, TG, LDL-C/HDL-C ratio, and non-HDL and on adiponectin are summarized in Table 5. There were no differences between the high-lycopene tomato and placebo groups in terms of changes in these parameters.

### 3.4. Effect of High-Lycopene Tomato on Serum Carotenoid Levels

To confirm the effects of high-lycopene tomato on serum carotenoid levels, changes in lycopene and β-carotene contents were evaluated (Table 6). The intake of high-lycopene tomato increased lycopene levels compared to the corresponding levels in the placebo group (ΔWeek 12: placebo, −24.2 ± 49.3 μg/dL; high-lycopene tomato, 22.7 ± 47.9 μg/dL; *p* < 0.001). In addition, β-carotene levels increased in the high-lycopene tomato group compared to those in the placebo group at week 12 (ΔWeek 12: placebo, −0.9 ± 13.6 μg/dL; high-lycopene tomato, 12.0 ± 24.5 μg/dL; *p* = 0.009).

### 3.5. Effect of High-Lycopene Tomato on Oxidative Markers

When the oxidative markers MDA-LDL, LOX index, sLOX-1, LAB, and LPO were examined, no statistically significant differences were found between the high-lycopene tomato and placebo groups (Table 7).

### 3.6. Effect of High-Lycopene Tomato on Fatigue and Stress

To determine the effects of high-lycopene tomato on fatigue and stress, we evaluated changes in saliva volume, CGA, and results of the VAS and POMS-2 questionnaires. There were no statistically significant differences between the high-lycopene tomato and placebo groups (data not shown).

### 3.7. Assessment of Dietary Nutrients among Subjects during the Study

To assess dietary nutrients, subjects completed the FFQg, which revealed no statistically significant differences in the intake of calories, proteins, lipids, carbohydrates, dietary fiber, and salt between the high-lycopene tomato and placebo groups (Appendix A), suggesting that dietary nutrients from meals did not affect the results of this trial.

### 3.8. Safety

To analyze the safety of high-lycopene tomato, we evaluated vital signs (SBP, DBP, and pulse rate), body composition (body weight [BW], BFR, and BMI), complete blood counts (WBC, RBC, Hb, Ht, and Plt), liver function (AST, ALT, γ-GTP, ALP, and LDH), renal function (BUN, CRE, and UA), and blood glucose profiles (FPG, HbA1c, and HOMA-IR) in the subjects (Appendix A), in addition to performing qualitative urinary assessments (pH, sugar, protein, occult blood, urobilinogen, and ketones) (data not shown). Mild adverse effects were observed in each group, with 20 adverse effects observed in the high-lycopene tomato group as follows: digestive symptoms (*n* = 9); malaise (*n* = 1); nasal discharge and cough (*n* = 1); numbness of limb (*n* = 1); skin symptoms (*n* = 2); abnormal value of prostate marker (*n* = 1); variation in γ-GTP value (*n* = 3); variation in UA value (*n* = 2). In the placebo group, 21 adverse effects were observed as follows: digestive symptoms (*n* = 3); injury (*n* = 1); stomatitis (*n* = 1); backache (*n* = 1); skin symptoms (*n* = 2); toothache (*n* = 1); urinary occult blood (*n* = 1); nasal symptoms (*n* = 3); variation in ALT (*n* = 1); variation in DBP (*n* = 1); variation in Hb (*n* = 1); variation in Ht (*n* = 1); variation in RBC (*n* = 1); variation in UA (*n* = 2); variation in γ-GTP (*n* = 1). All subjects displayed mild symptoms and recovered within a few days. In addition, no symptoms were associated with variations in laboratory test values; the principal investigator inferred that there were no side effects and no serious adverse effects in this trial. Thus, the intake of high-lycopene tomato had no or minimal unfavorable effects, even at 200 g/day (as a raw tomato).

## 4. Discussion

In this clinical trial, we assessed the effect of daily intake of high-lycopene tomato for 12 weeks on lipid profiles in Japanese subjects with LDL-C levels ≥120 mg/dL and <160 mg/dL. The intake of high-lycopene tomato increased lycopene levels in subjects administered this food compared to those in the placebo group. In addition, LDL-C, the primary outcome, was improved in the high-lycopene tomato group. 

A previous meta-analysis demonstrated that LDL-C decreases when more than 25 mg per day of lycopene is ingested [17]. The biological mechanism was associated with a reduction in 3-hydroxy-3-methyl-glutaryl-coenzyme A (HMG-CoA) reductase activity in the liver [4], activation of LDL-receptors [5], and increased expression of the ABCA1 transporter gene, the key component of HDL-C production [6]. Likewise, these results were observed in our clinical trial using semidried tomato. To our knowledge, this is the first study to investigate the effect of semidried tomato, which is similar to raw tomato, on lipid profiles based on a double-blind, placebo-controlled trial.

The intake of high-lycopene tomato increased the level of lycopene compared to the corresponding level in the placebo group. In the present clinical trial, subjects were asked to refrain from ingesting tomatoes and processed foods containing tomatoes, and to avoid drinking vegetable juices; thus, the test food was thought to be the only tomato derivative consumed. Serum lycopene was decreased in the placebo group. In addition, serum β-carotene was increased in the high-lycopene tomato group. The level of β-carotene contained in the active test food was less than 4 mg/day, but 20 mg/day β-carotene was previously found to be required for an effect on lipid metabolism [18]. These results suggest that the improvement of LDL-C in our clinical trial was mainly due to the effect of lycopene. 

MDA-LDL and LPO were not affected by actively consuming the test food; however, based on the exploratory analysis, a positive correlation was observed between the change in LDL-C and the change in MDA-LDL in the active test food group (Pearson correlation coefficient, r = 0.444, *p* = 0.010). In addition, subjects with improved LDL-C levels due to intake of the active test food were also suggested to have improved MDA-LDL. Regarding the antioxidant activity of carotenoids, lycopene has been reported to possess the strongest singlet oxygen scavenging ability among the eight carotenoids, as measured by the singlet oxygen absorption capacity method [19], and some researchers have found that lycopene and tomato display antioxidant effects [20,21,22]. However, other reports suggest that ingesting lycopene does not affect oxidative markers [23,24], although they assessed various oxidative markers such as MDA-LDL, antioxidant capacity, 8-iso-PGF, and antioxidant enzymes, and during the intake period, there was apparent variation in the antioxidant effect. These findings suggest that a re-evaluation of oxidant markers and the intake period is required. 

The LOX index, sLOX-1, and LAB, did not significantly differ between the active test food and placebo groups. The LOX index is a biomarker for the early risk of arteriosclerosis, cerebral infarction, and myocardial infarction [25,26], and is calculated based on LAB and sLOX-1. sLOX-1 recognizes not only oxLDL, but also other atherogenic lipoproteins, platelets, leukocytes, and CRP [27]. The antioxidant effects of lycopene comprise the main mechanism associated with singlet oxygen scavenging. Therefore, lycopene might be ineffective against LAB or sLOX-1, which are the products of the peroxidation reaction. As subjects in our trials were healthy and the study period was too short to investigate the effect of lycopene on the LOX index, additional studies with a longer intake period are required to reveal the effect of tomato on the risk of arteriosclerosis.

One limitation of this study was the ingestion period. Hence, it is imperative to reconsider the effects of long-term ingestion using other evaluation outcomes. As the absorption of lycopene is understood to be influenced by food processing, cooking, and meal composition such as type and amount of fat, subjects were asked to avoid cooking the test food to eliminate such effects on absorption due to differences in cooking methods. Thus, it is also necessary to examine differences in the effect of high-lycopene tomato based on various cooking methods. In addition, a previous study suggested that the plasma lycopene half-life is approximately six days, based on results of the continuous ingestion of 20 mg lycopene for eight days [28]. We thus need to consider the effect of high-lycopene tomato after the end of the ingestion period.

## 5. Conclusions

In this 12-week randomized, double-blind, placebo-controlled, parallel-group comparative study, the intake of high-lycopene semidried tomato, PR-7, reduced LDL-C and was confirmed to be safe at a dosage of 200 g/day (as raw tomato). Tomato is an important component of the diet worldwide, and our findings support the health benefits, especially with respect to lipid metabolism, of consuming tomatoes rich in lycopene.

## Figures and Tables

**Figure 1 nutrients-11-01177-f001:**
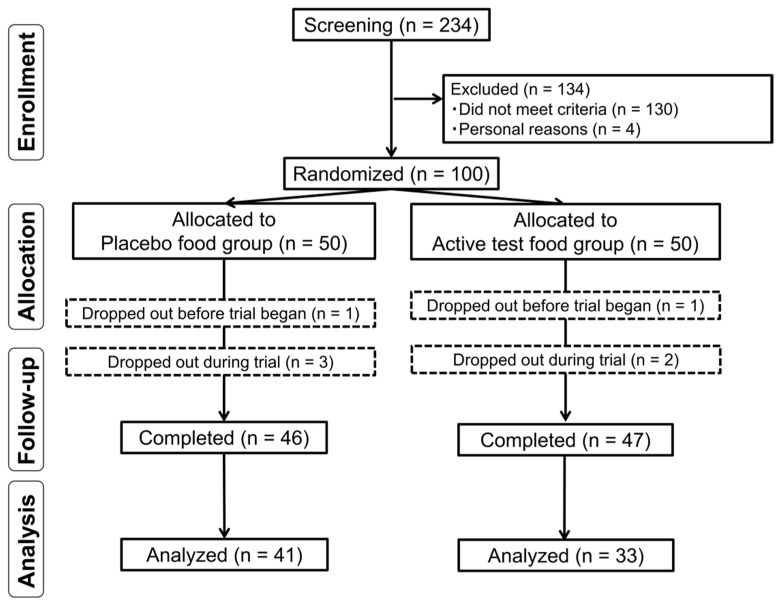
Subject selection.

**Figure 2 nutrients-11-01177-f002:**
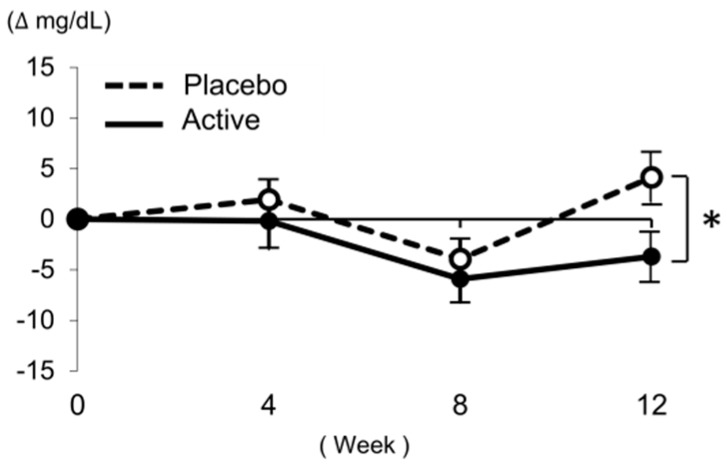
Changes in low-density lipoprotein cholesterol (LDL-C) in all subjects. A Student’s *t*-test was conducted for data analysis. Statistical significance: * *p* < 0.05 vs. placebo group. Values are presented as the mean ± standard error. Number of subjects (*n*): placebo: *n* = 41, active: *n* = 33.

**Table 1 nutrients-11-01177-t001:** Clinical trial schedule.

Item	Guidance & Agreement	Screening	Randomization	Test Food-Intake Period			
				Week 0	Week 4	Week 8	Week 12
Visit	Visit 1		–	Visit 2	Visit 3	Visit 4	Visit 5
Date	20–23 April 2018		21 May 2018	9–11 June 2018	7–9 July 2018	4–6 August 2018	1–3 September 2018
Medical interview		●		●	●	●	●
Vital sign measurement		●		●	●	●	●
Body composition measurement		●		●	●	●	●
Blood sampling		●		●	●	●	●
Urinary and salivary test				●	●	●	●
VAS questionnaire and POMS-2				●	●	●	●
Food Frequency Questionnaire				●	●	●	●

POMS-2: Profile of Mood States Second Edition; ●: performed.

**Table 2 nutrients-11-01177-t002:** Inclusion and exclusion criteria.

Inclusion criteria	1. Age, ≥30 years and <70 years old
	2. LDL-C, ≥120 mg/dL and <160 mg/dL
Main exclusion criteria	1. Subjects who participated in the pilot study
	2. Subjects who usually do not consume raw tomatoes
	3. Subjects who usually consume tomato juice
	4. Subjects under physician’s advice, treatment, and/or medication for dyslipidemia and/or diabetes
	5. Subjects with a BMI ≥30 kg/m^2^
	6. Subjects with familial hypercholesterolemia
	7. Subjects with serious cerebrovascular, cardiac, hepatic, renal, gastrointestinal diseases, and/or affected by infectious diseases requiring reports to the authorities
	8. Subjects with a major surgical history relevant to the digestive system, such as gastrectomy, gastrorrhaphy, enterectomy, etc.
	9. Subjects with unusually high and/or low blood pressure and/or abnormal hematological data
	10. Subjects with severe anemia
	11. Pre- or post-menopausal women complaining of obvious physical changes
	12. Subjects at risk of allergic reactions to drugs or foods especially due to tomato, Japanese cedar, Japanese cypress, or grass
	13. Subjects who regularly take medications, functional foods, and/or supplements, which would affect blood lipid and/or glucose metabolism
	14. Alcohol addicts or subjects with an eating disorder
	15. Subjects who donated either 400 mL of whole blood within 16 weeks (women) or 12 weeks (men), 200 mL of whole blood within 4 weeks (men and women), or blood components within 2 weeks (men and women) prior to the current study.
	16. Pregnant or lactating women or women who expect to be pregnant during this study
	17. Subjects who currently participate in other clinical trials or have participated in a trial within the last 4 weeks prior to the current study
	18. Any other medical and/or health reasons unfavorable to participation in the current study, as judged by the principal investigator

**Table 3 nutrients-11-01177-t003:** Nutrient compositions of the active test and placebo foods based on daily consumption.

Nutrient	Active Test Food	Placebo Food
Calories (kcal)	57.5	45.7
Water (g)	33.2	36.5
Proteins (g)	2.1	2.1
Lipids (g)	0.4	0.4
Carbohydrates (g)	12.8	9.8
Ash (g)	1.7	1.2
Total fiber (g)	2.6	2.6
Sodium (mg)	3.1	2.3
Lycopene (mg)	22.0–27.8	n.d.
β-Carotene (mg)	2.8-3.3	n.d.

n.d.: not detected.

**Table 4 nutrients-11-01177-t004:** Characteristics of subjects and intake rates of test foods in the active test food and placebo groups.

Characteristic	Placebo	Active	*p*
Subjects, *n*	41	33	-
Male, *n*	17	8	0.14
Age, years	53.9 ± 9.1	53.7 ± 7.8	0.91
Height, cm	162.7 ± 8.8	160.4 ± 8.3	0.27
Body mass index, kg/m^2^	21.7 ± 2.8	22.1 ± 3.0	0.59
LDL cholesterol, mg/dL	137.3 ± 12.4	139.2 ± 12.5	0.51
Intake rate, %	99.8 ± 0.5	99.9 ± 0.4	0.45

**Table 5 nutrients-11-01177-t005:** Lipid profiles and adiponectin levels for the placebo and active food test groups.

Variable		Week 0	ΔWeek 4	ΔWeek 8	ΔWeek 12	Time × Food Interaction, *p* ^b^
LDL-C (mg/dL)	Placebo (*n* = 41)	133.4 ± 15.6	1.9 ± 12.1	−4.0 ± 12.4	4.1 ± 15.7	0.13
	Active (*n* = 33)	140.2 ± 16.9	−0.2 ± 15.5	−5.9 ± 13.0	−3.7 ± 13.8	
	*p* ^a^	0.078	0.514	0.511	0.027 *	
LDL-C subjects whose LDL-C was 120–139 mg/dL (mg/dL)	Placebo (*n* = 41)	122.1±10.1	1.8±8.5	-2.0±10.6	4.3±15.1	0.048 *
	Active (*n* = 33)	127.8±10.1	3.2±11.9	-6.2±7.8	-5.1±9.5	
	*p* ^a^	0.100	0.686	0.180	0.030 *	
LDL-C subjects whose LDL-C was 140–159 mg/dL (mg/dL)	Placebo (*n* = 41)	143.2 ± 12.6	2.0 ± 14.9	−5.7 ± 13.9	4.0 ± 16.6	0.354
	Active (*n* = 33)	153.4 ± 11.8	−3.9 ± 18.3	−5.6 ± 17.2	−2.2 ± 17.5	
	*p* ^a^	0.016 *	0.286	0.981	0.276	
TC (mg/dL)	Placebo (*n* = 41)	222.5 ± 23.9	0.9 ± 15.1	−1.5 ± 18.0	7.9 ± 21.7	0.20
	Active (*n* = 33)	231.8 ± 24.6	−1.6 ± 18.5	−1.3 ± 17.0	1.2 ± 19.1	
	*p* ^a^	0.10	0.51	0.96	0.17	
HDL-C (mg/dL)	Placebo (*n* = 41)	75.2 ± 17.5	−1.0 ± 5.8	−1.9 ± 7.5	−0.6 ± 8.4	0.76
	Active (*n* = 33)	80.7 ± 17.4	−3.0 ± 7.3	−3.0 ± 6.5	−2.7 ± 7.8	
	*p* ^a^	0.18	0.21	0.49	0.27	
TG (mg/dL)	Placebo (*n* = 41)	86.6 ± 30.7	−2.4±25.5	−3.7 ± 28.5	0.4 ± 24.8	0.83
	Active (*n* = 33)	80.8 ± 32.8	−3.2±24.1	−0.9 ± 20.6	3.2 ± 31.2	
	*p* ^a^	0.43	0.89	0.64	0.66	
LDL-C/HDL-C ratio	Placebo (*n* = 41)	1.9 ± 0.4	0.1 ± 0.2	0.0 ± 0.2	0.1 ± 0.2	0.24
	Active (*n* = 33)	1.8 ± 0.4	0.1 ± 0.2	0.0 ± 0.2	0.0 ± 0.2	
	*p* ^a^	0.67	0.67	0.72	0.33	
non-HDL (mg/dL)	Placebo (*n* = 41)	147.3 ± 15.9	1.9 ± 14.0	0.4 ± 14.1	8.5 ± 17.0	0.20
	Active (*n* = 33)	151.2 ± 16.9	1.4 ± 14.1	1.8 ± 13.2	3.9 ± 15.6	
	*p* ^a^	0.32	0.87	0.67	0.24	
Adiponectin (μg/mL)	Placebo (*n* = 41)	10.9 ± 5.9	−0.3 ± 1.1	−0.6 ± 1.3	−0.5 ± 1.2	0.005 **
	Active (*n* = 33)	13.6 ± 7.1	0.1 ± 1.1	−0.6 ± 1.4	−1.0 ± 1.0	
	*p* ^a^	0.085	0.17	0.97	0.053	

Values are shown as the mean ± standard deviation. *p*
^a^: Student’s *t*-test was performed. *p*
^b^: repeated measures of analysis of variance was performed. * *p* < 0.05, ** *p* < 0.01 vs. placebo group. ΔWeek 4: changes in values from baseline to week 4; ΔWeek 8: changes in values from baseline to week 8; ΔWeek 12: changes in values from baseline to week 12; LDL-C: low-density lipoprotein cholesterol; TC: total cholesterol; HDL-C: high-density lipoprotein cholesterol; TG: triglycerides; LDL-C/HDL-C ratio: LDL cholesterol/HDL cholesterol ratio; non-HDL: non-low-density lipoprotein cholesterol.

**Table 6 nutrients-11-01177-t006:** Serum carotenoid levels for the placebo and active food test groups.

Variable		Week 0	ΔWeek 4	ΔWeek 8	ΔWeek 12	Time × Food Interaction, *p* ^b^
Lycopene (μg/dL)	Placebo (*n* = 41)	75.2 ± 45.9	−26.1 ± 36.2	−20.5 ± 47.8	−24.2 ± 49.3	0.64
	Active (*n* = 33)	85.9 ± 53.0	14.1 ± 49.0	25.4 ± 42.0	22.7 ± 47.9	
	*p* ^a^	0.36	*p* < 0.001 **	*p* < 0.001 **	*p* < 0.001 **	
β-carotene (μg/dL)	Placebo (*n* = 41)	37.6 ± 29.2	−2.2 ± 10.6	−0.3 ± 13.9	−0.9 ± 13.6	0.45
	Active (*n* = 33)	51.8 ± 45	10.9 ± 20.9	16.2 ± 25.1	12.0 ± 24.5	
	*p* ^a^	0.13	0.002 **	0.001 **	0.009 **	

Values are shown as the mean ± standard deviation. *p*
^a^: Student’s *t*-test was performed. *p*
^b^: repeated measures of analysis of variance was performed. ** *p* < 0.01 vs. placebo group. ΔWeek 4: changes in values from baseline to week 4; ΔWeek 8: changes in values from baseline to week 8; ΔWeek 12: changes in value from baseline to week 12.

**Table 7 nutrients-11-01177-t007:** Oxidative marker levels for the placebo and active food test groups.

Variable		Week 0	ΔWeek 4	ΔWeek 8	ΔWeek 12	Time × Food Interaction, *p* ^b^
MDA-LDL (U/L)	Placebo (*n* = 41)	156.6 ± 47.5	−45.6 ± 49.5	−19.5 ± 47.9	6.8 ± 54.5	0.94
	Active (*n* = 33)	151.5 ± 45.2	−41.3 ± 32.7	−14.9 ± 48.7	7.8 ± 46.9	
	*p* ^a^	0.65	0.65	0.68	0.94	
LOX-index	Placebo (*n* = 41)	459.7 ± 514.9	−141.2 ± 299.2	−43.8 ± 287.2	698.5 ± 3527.1	0.41
	Active (*n* = 33)	303.1 ± 118.4	−40.8 ± 165.7	45.2 ± 250.3	278.8 ± 311.4	
	*p* ^a^	0.066	0.089	0.17	0.50	
sLOX-1 (pg/mL)	Placebo (*n* = 41)	472.3 ± 511.1	11.3 ± 243.2	19.3 ± 176.2	675.0 ± 3688.7	0.39
	Active (*n* = 33)	291.7 ± 127.1	127.6 ± 333.5	103.9 ± 196.3	226.0 ± 242.1	
	*p* ^a^	0.034 *	0.087	0.055	0.49	
LAB (μg·cs/mL)	Placebo (*n* = 41)	1.1 ± 0.3	−0.4 ± 0.3	−0.2 ± 0.3	0.0 ± 0.3	0.23
	Active (*n* = 33)	1.1 ± 0.2	−0.4 ± 0.3	−0.2 ± 0.3	0.1 ± 0.3	
	*p* ^a^	0.67	0.63	0.63	0.48	
LPO (nmol/mL)	Placebo (*n* = 41)	4.2 ± 0.6	−1.1 ± 0.6	−0.7 ± 0.6	−0.5 ± 0.6	0.34
	Active (*n* = 33)	4.1 ± 0.5	−1.2 ± 0.6	−0.7 ± 0.5	−0.5 ± 0.4	
	*p* ^a^	0.83	0.32	0.87	0.93	

Values are shown as the mean ± standard deviation. *p*
^a^: Student’s *t*-test was performed. *p*
^b^: repeated measures of analysis of variance was performed. * *p* < 0.05 vs. placebo group. ΔWeek 4: changes in values from baseline to week 4; ΔWeek 8: changes in values from baseline to week 8; ΔWeek 12: changes in value from baseline to week 12; MDA-LDL: malondialdehyde LDL-C; sLOX-1: soluble lectin-like oxidized LDL receptor-1; LAB: LOX-1 ligand containing apolipoprotein B; LPO: lipid peroxidases.

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
