# Peer review of "Effect of 12-Week Daily Intake of the High-Lycopene Tomato (Solanum Lycopersicum), a Variety Named “PR-7”, on Lipid Metabolism: A Randomized, Double-Blind, Placebo-Controlled, Parallel-Group Study"

_nutrients, 2019, doi:10.3390/nu11051177_

Round 1
Reviewer 1 Report
The authors evaluated the impact of a high-lycopene tomato product on serum lipids. There are numerous questions that must be addressed.
Abstract: It must be stated that the analyses are based upon 74 persons, not the 100 who were enrolled.
Line 59 - 61. What does "required amount" refer to? Required for what?
Line 62. "proven" is too strong of a term and should be removed.
What does "semi-dried" mean? Describe the consistency of the product.
A major issue is - how would one "blind" the subjects and the researchers if one product was extremely red due to the high lycopene and the other was very pale because of the lack of lycopene?
Lines 161 - 165. Please provide a reference for the FFQ. Was it validated? How many days was it run?
Methods - Safety evaluation. No details are provided on how this was performed. In addition, no data are show for the results. Both of these are important.
Lines 194 - 211. It is not correct that "Finally, 93 subjects completed the trial..." if only 74 had data evaluated. Why were 5 subjects lost in the "placebo" and 14 in the "Active" group for analysis. What were the reasons for lack of analyses?
In Table 4, it is indicated that N=74 but in other results, it not stated what the "N" was for data analysis. For example, for Figure #2, what was the "N" for each time point? The number of subjects analyzed for all data needs to be indicated.
Table 5 - LDL - C week zero levels are not the same as shown in Table 4.
Why wasn't the beta carotene levels in the two products included in Table 3? This is important.
Lines 325 - 330. This reviewer fails to see the value of the discussion and speculation regarding glutamic acid. Remove.
Supplemental Table 1. Please indicate that these are daily intakes. Please indicate how many days were evaluated for the FFQs.
Author Response
We wish to express our strong appreciation to you for your insightful comments on our paper. The attached file is a response to the questions and comments delivered in your advice.
Point 1: Abstract: It must be stated that the analyses are based upon 74 persons, not the 100 who were enrolled.
Response 1: Thank you for your comment. We corrected it (Abstract).
Point 2: Line 59 - 61. What does "required amount" refer to? Required for what?
Response 2: Thank you for your comment. We have meant required amount to improve to cholesterol. We clarified these sentences. (Line 61-63)
Point 3: Line 62. "proven" is too strong of a term and should be removed.
Response 3: Thank you for your suggestion. We changed it to “shown”. (Line 64)
Point 4: What does "semi-dried" mean? Describe the consistency of the product.
Response 4: Thank you for providing these insights. Semidried tomato contained 22 % water content, not was completely dried foods.
We add the sentence. (Line 68)
Point 5: A major issue is - how would one "blind" the subjects and the researchers if
one product was extremely red due to the high lycopene and the other was
very pale because of the lack of lycopene?
Response 5: Thank you for your important suggestion. As you point, color was different between two tomatoes. However, the food was packaged in an opacity aluminum pack, so this difference was not recognized from appearance. In addition, the test food manager controlled the subjects by the food-controlled number printed on the package, and the information which number is active or placebo had been closed until all analytical data were collected. Moreover, active and placebo tomato had not sold in japan during the study period, therefor subjects never could not confirm which tomatoes they ingestion. We add the sentence. (Line 118-119)
Point 6: Lines 161 - 165. Please provide a reference for the FFQ. Was it validated?
How many days was it run?
Response 6: Thank you for your suggestion. FFQg was validated by Japanese researchers.
Subjects reported the weekly amount and frequency of food intake over the past month at each visit. We add the sentence and reference (Line174-178).
Point 7: Methods - Safety evaluation. No details are provided on how this was
performed. In addition, no data are show for the results. Both of these are
important.
Response 7: Thank you for your suggestion. We add these method and result section. (Line
180-186 and Line 301-308)
Point 8: Lines 194 - 211. It is not correct that "Finally, 93 subjects completed the
trial..." if only 74 had data evaluated. Why were 5 subjects lost in the
"placebo" and 14 in the "Active" group for analysis. What were the reasons
for lack of analyses?
Response 8: Thank you for your comment. We described the number of efficacy analysis and reason for exclusion (Line 227-231).
Point 9: In Table 4, it is indicated that N=74 but in other results, it not stated what the
"N" was for data analysis. For example, for Figure #2, what was the "N" for
each time point? The number of subjects analyzed for all data needs to be
indicated.
Response 9: Thank you for your comment. We add the number of subjects. (Figure 2, Table 5-7, Supplementary Table 1-2)
Point 10: Table 5 - LDL - C week zero levels are not the same as shown in Table 4.
Response 10: Thank you for your suggestion. The levels in Table4 are at visit 1 (screening).
On the other hand, the levels in Table5 are at visit 2 (week 0). We add the sentence. (Line 231)
Point 11: Why wasn't the beta carotene levels in the two products included in Table 3?
This is important.
Response 11: Thank you for your comment. We added data. (Table3)
Point 12: Lines 325 - 330. This reviewer fails to see the value of the discussion and
speculation regarding glutamic acid. Remove.
Response 12: Thank you for your suggestion. We deleted this section.
Point 13: Supplemental Table 1. Please indicate that these are daily intakes. Please
indicate how many days were evaluated for the FFQs.
Response 13: Thank you for your suggestion. We changed the table title. (Supplemental
Table 1)
Reviewer 2 Report
Comments for Authors:
The manuscript entitled “Effect of Daily Intake of the High-Lycopene Tomato (Solanum Lycopersicum), variety named “PR-7”, for 12 weeks on Lipid Metabolism: A Randomized, Double-Blind, Placebo-Controlled, Parallel-Group Study” by Nishimura and colleagues sought to assess the beneficial effect of Lycopene-rich tomato PR-7 intake on lipid metabolism, specifically LDL-C levels. Overall, this is a well-written manuscript, and the data are scientifically sound with appropriate controls. The authors have also taken steps to assess the safety of the PR-7 intake and the efficacy of high-lycopene tomato on fatigue and stress.
This is an interesting study, and the findings may have potential translational and marketing implications. Below are suggestions to strengthen the study further:
Major comments:
1. The authors should justify the standard of selecting subjects with LDL-C levels >= 120 to < 160mg/dL according to current clinical diagnostic test standard. Besides, the author separated the subjects to groups of LDL-C 120-129 and 140-159 mg/dL for differential effects in table 5. Is 120-160 considered as healthy, borderline, or high in this study? Please note that different countries might use different diagnostic standard or unit.
2. In Table 3, the total Calories (kcal) and Carbohydrates (g) levels are ~20% lower in the Placebo food compared to Active test food? The author should clarify that if this is a significant difference in this study and why?
3. In Figure 2, there appears to be a consistent decrease in both Placebo and Active group at Week 8 compared to week 0. Is it expected or is it just background noise due to different testing time?
Minor comments:
1. In Figure 2, the significant difference was only shown at one time point, week 12. The author already acknowledged the limitation of the ingestion period, and it is imperative to reconsider the effects of long-term ingestion. However, I think the author should also comment on the potential of the lasting benefits from the 12-week PR-7 consumption on LDL-C levels even after withdrawn.
2. In the title, there should a "a" before "variety named PR-7".
3. In the checklist, the authors have no explanation on why the trail ended or stopped.
Author Response
We wish to express our strong appreciation to you for your insightful comments on our paper.
We wish to express our strong appreciation to you for your insightful and kind comments on our paper. The following is a response to the questions and comments delivered in your advice.
Point 1: The authors should justify the standard of selecting subjects with LDL-C levels >= 120 to < 160mg/dL according to current clinical diagnostic test standard. Besides, the author separated the subjects to groups of LDL-C 120-139 and 140-159 mg/dL for differential effects in table 5. Is 120-160 considered as healthy, borderline, or high in this study? Please note that different countries might use different diagnostic standard or unit.
Response 1: Thank you for your suggestion. We have referred to “The Guideline for
Prevention of Arteriosclerosis Disease” published by Japan Atherosclerosis Society; borderline high LDL-C: 120-139 mg/dl; mild hypercholesterolemia:140-159 mg/dl, and > 160 mg/dl; need the treatment. We added it. (Line 109-112 and 202-203)
Point 2: In Table 3, the total Calories (kcal) and Carbohydrates (g) levels are ~20%
lower in the Placebo food compared to Active test food? The author should
clarify that if this is a significant difference in this study and why?
Response 2: Thank you for providing these insights. Supplementary Table1 shows that subjects constantly intake calories of 1,700 kcal and carbohydrates of 200 g from their daily diet. Compared to the daily intake of nutrition from the diet, this difference was insignificant; therefore, we consider that this would not affect our data. We add it (Line 132-136.)
Point 3: In Figure 2, there appears to be a consistent decrease in both Placebo
and Active group at Week 8 compared to week 0. Is it expected or is it just
background noise due to different testing time?
Response 3: Thank you for your important suggestion. We speculated that the decrease at week 8 might be just placebo effect or caused by the something component contained in both tomatoes. However, at week 12, LDL-C decreased following ingestion of only active food; therefore we thought the improvement of LDL-C at week 12 was mainly due to the effect of lycopene.
Point 4: In Figure 2, the significant difference was only shown at one time point,
week 12. The author already acknowledged the limitation of the ingestion
period, and it is imperative to reconsider the effects of long-term ingestion.
However, I think the author should also comment on the potential of the
lasting benefits from the 12-week PR-7 consumption on LDL-C levels even
after withdrawn.
Response 4: Thank you for providing these insights. We need to consider the about the effect of high-lycopene tomato after the end of intake. We add it in limitation section (Line 361-374).
Point 5: In the title, there should a "a" before "variety named PR-7".
Response 5: Thank you for your comment. We changed title and others. (Title, abstract, Line 60, Line 124)
Point 6: In the checklist, the authors have no explanation on why the trail ended or
stopped.
Response 6: Thank you for your comment. Our study was completed as planned because
there was no reason to stop the study. We added it (Line 223-225), and corrected the checklist.
Round 2
Reviewer 1 Report
The authors have addressed the reviewer's concerns.